# Heterologous Expression of Poplar *WRKY18/35* Paralogs in Arabidopsis Reveals Their Antagonistic Regulation on Pathogen Resistance and Abiotic Stress Tolerance via Variable Hormonal Pathways

**DOI:** 10.3390/ijms21155440

**Published:** 2020-07-30

**Authors:** Li Guo, Chaofeng Li, Yuanzhong Jiang, Keming Luo, Changzheng Xu

**Affiliations:** 1Chongqing Key Laboratory of Plant Resource Conservation and Germplasm Innovation, School of Life Sciences, Southwest University, Chongqing 400715, China; gl1231@yeah.net (L.G.); lichaofeng.1987@163.com (C.L.); jyz88623@126.com (Y.J.); 2Crop Functional Genomics, Institute of Crop Science and Resource Conservation (INRES), University of Bonn, 53113 Bonn, Germany; 3Asian Natural Environmental Science Center, The University of Tokyo, 1-1-8 Midori-cho, Nishitokyo, Tokyo 188-0002, Japan; 4Key Laboratory for Bio-resources and Eco-environment of Ministry of Education, College of Life Science, Sichuan University, Chengdu 610065, China

**Keywords:** *PtrWRKY18*, *PtrWRKY35*, multifunctional, pathogen resistance, water-deficit

## Abstract

WRKY transcription factors (WRKY TFs) are one of the largest protein families in plants, and most of them play vital roles in response to biotic and abiotic stresses by regulating related signaling pathways. In this study, we isolated two WRKY TF genes *PtrWRKY18* and *PtrWRKY35* from *Populus*
*trichocarpa* and overexpressed them in Arabidopsis. Expression pattern analyses showed that *PtrWRKY18* and *PtrWRKY35* respond to salicylic acid (SA), methyl JA (MeJA), abscisic acid (ABA), *B. cinereal*, and *P. syringae* treatment. The transgenic plants conferred higher *B. cinerea* tolerance than wild-type (WT) plants, and real-time quantitative (qRT)-PCR assays showed that *PR3* and *PDF1.2* had higher expression levels in transgenic plants, which was consistent with their tolerance to *B. cinereal.* The transgenic plants showed lower *P. syringae* tolerance than WT plants, and qRT-PCR analysis (*PR1*, *PR2*, and *NPR1*) also corresponded to this phenotype. Germination rate and root analysis showed that the transgenic plants are less sensitive to ABA, which leads to the reduced tolerance to osmotic stress and the increase of the death ratio and stomatal aperture. Compared with WT plants, a series of ABA-related genes *(RD29A*, *ABO3*, *ABI4*, *ABI5*, and *DREB1A*) were significantly down-regulated in *PtrWRKY18* and *PtrWRKY35* overexpression plants. All of these results demonstrated that the two WRKY TFs are multifunctional transcription factors in plant resistance.

## 1. Introduction 

Thanks to their sessile lifestyle, high plants have to face various biotic and abiotic stresses [1]. To cope with these challenges, plants have evolved sophisticated mechanisms to perceive these environmental stresses and respond optimally [2]. The activation of defense or acclimation machinery plays important roles in preventing further damages to the entire plant. The signals that participate in plant responses are usually divided into two types: fast-moving signals that react within minutes, such as methyl JA (MeJA) or methyl SA (MeSA), and slow-moving signals that need several hours to transport and respond to changes, such as jasmonic acid (JA), salicylic acid (SA), or azelaic acid [3].

Recognition and transduction of stress signals to activate plant responses and regulation of stress-resistant genes are the key steps for enabling stress tolerance in plants. The stress-resistant genes are mainly induced at the transcriptional level, and many transcription factor (TF) families such as WRKY, AP2 (APETALA2)/ERF (ethylene responsive factor), and NAC (NAM, ATAF1/2, CUC1/2) play crucial roles in activating the expression of many stress-resistant genes during diverse biotic and abiotic stress responses [4].

WRKY transcription factors (WRKYs) belong to one of the largest TF families in plants, which are named by the highly conserved DNA-binding region WRKY domain (the WRKYGQK motifs at the N-terminal and a zinc finger motif at the C-terminus) [5]. Depending on the numbers of WRKY domain and the features of zinc-finger motif, WRKYs are divided into three groups [6,7]: Group I (containing two WRKY domains and one zinc-finger-like motif C2H2), Group II (containing one WRKY domain and one C2H2 zinc-finger-like motif), and Group III (containing one WRKY domain and one C2HC zinc-finger-like motif [8].

Increasing evidence demonstrates that WRKYs play vital roles in pathogen defense, and are regulated by elicitors such as wounding, SA, and JA. For example, *CaWRKY2* and *PtrWRKY40* play roles in resistance to pathogens [9,10]. *AtWRKY25* is involved in plant defense against *Pseudomonas syringae* (*P. syringae*) [11], while *AtWRKY3* and *AtWRKY4* enhance the defense against *Botrytis cinerea* (*B. cinereal*) [12]. WRKYs also participate in plant responses to abiotic stresses and abscisic acid (ABA) signaling [13]. In Arabidopsis, the knock-out of *ABO3* encoding a WRKY transcription factor decreased the ABA sensitivity, but increased the sensitivity to drought stress [14]. *AtWRKY18*, *AtWRKY40*, and *AtWRKY60* have been found to not only function in plant responses to pathogen defense, but also regulate the response to drought stress [15,16]. In rice, the expressions of *OsWRKY24*, *OsWRKY51*, *OsWRKY71*, and *OsWRKY72* are induced by ABA [17]. The soybean *GmWRKY13*, *GmWRKY21*, and *GmWRKY54* improve the abiotic stress tolerance in transgenic Arabidopsis. Moreover, WRKYs also play roles in seed development and germination [18], leaf senescence [19], and secondary metabolism [20].

We previously identified paralogous *PtrWRKY18* and *PtrWRKY35* encoding the homologs of Arabidopsis *WRKY18*, *WRKY40*, and *WRKY60* in poplar [21]. Functional characterization revealed that both *WRKY* paralogs redundantly regulate the defense against biotrophic pathogen and SA-mediated signaling pathway [22]. In this study, we heterologously expressed the poplar *WRKY18* and *WRKY35* in Arabidopsis, and found their differential regulation on pathogen resistance and abiotic stress tolerance. Our results indicated that these WRKY transcription factors modulate different hormonal signaling to confer multiple biotic and abiotic stress responses.

## 2. Results

### 2.1. The Expression of PtrWRKY18 and PtrWRKY35 Is Induced by Various Hormones and Biotic Stresses

To comprehensively understand the role of *PtrWRKY18* and *PtrWRKY35* in stress response, the GUS (β-glucuronidase) reporter lines driven by their promoter fragments were firstly generated in Arabidopsis. The homozygous transgenic plants with single copy insertion were exposed to various hormones and fungal pathogens, and determined by histological staining (Figure 1). The expression of *PtrWRKY18* and *PtrWRKY35* was significantly induced by *P. syringae* and *B. cinerea*, two pathogen species, as well as by SA and JA, two hormones involved in pathogen resistance (Figure 1A). In contrast, we found that the expression of both *WRKY* paralogs was largely repressed under ABA treatment for 4 h (Figure 1B), which was validated by time-course quantification of GUS activity (Figure 1C). The differential responses of the WRKY expression to hormones and pathogens implicate their variable roles in stress resistance.

### 2.2. PtrWRKY18 and PtrWRKY35 Play Positive Roles in Resistance Against B. cinerea

To investigate the roles of *PtrWRKY18* and *PtrWRKY35*, their overexpression vectors were transformed into WT Arabidopsis. Two independent homozygous lines containing a single insert with high *PtrWRKY18* (L2 and L4) or *PtrWRKY35* (L6 and L9) expression levels were selected by assays of semi-RT-PCR (Figure 2A) and qRT-PCR (Figure 2B), respectively, and used for further analyses.

The persistent expression of disease resistance genes can affect plant growth and development [23], hence we observed and compared the growth of WT and transgenic Arabidopsis. There was no significant difference between the two-week-old transgenic and WT seedlings in culture dishes (Figure 2B). Further, the seedlings were transferred into soil, and the 30-day-old WT seedlings were found to be slightly larger than the transgenic seedlings (Figure 2C).

In Arabidopsis, *AtWRKY18*, *AtWRKY40*, and *AtWRKY60* have been proven to function in defense of plants against pathogens, such as *B. cinerea* and *P. syringae* [15]. In order to verify if *PtrWRKY18* or *PtrWRKY35* have similar functions, the transgenic plants overexpressing *PtrWRKY18* or *PtrWRKY35* were sprayed with the spore suspending of *B. cinerea.* As shown in Figure 2D, the leaves of transgenic Arabidopsis overexpressing *PtrWRKY18* and PtrWRKY35 had a lesser necrosis area than that in WT leaves. To determine the growth status of *B. cinerea* in different genotypes of *Arabidopsis thaliana*, qRT-PCR was used to analyze the expression levels of *B. cinerea actin* gene. The results showed that the *B. cinerea* actin gene had lower expression levels in the transgenic plants (Figure 2E). Therefore, we deem that overexpression of *PtrWRKY18* or *PtrWRKY35* could improve the resistance to *B. cinerea* in Arabidopsis. In general, plant resistance to herbivorous insects or necrotrophic pathogens (such as *B. cinerea*) is achieved by activation of the JA signaling pathway. qRT-PCR showed that transcript levels of *PR3* and *PDF1.2* (Figure 2F), two marker genes in the JA signaling pathway [10], were significantly upregulated in transgenic plants, while *VSP2* (Figure 2F) showed no difference between WT and transgenic plants.

These results demonstrated that *PtrWRKY18* and *PtrWRKY35* have positive roles in JA-mediated signaling against *B. cinerea* in Arabidopsis.

### 2.3. PtrWRKY18 and PtrWRKY35 Play Negative Roles in Resistance Against P. syringae

In our previous studies, we found *PtrWRKY18* and *PtrWRKY35* function in SA signaling pathway and regulated resistance to the biotrophic pathogen *Melampsora* in *P. tomentosa* [22]. To explore whether heterologous expression of *PtrWRKY18* and *PtrWRKY35* in Arabidopsis affects their function in the SA signaling pathway, *P. syringae* were inoculated on the leaves to observe their development. As shown in Figure 3A,B, after *P. syringae* inoculation, the leaves of transgenic plants had more severe chlorosis spot symptoms than WT plants. Chlorophyll contents measurement showed that the contents of chlorophyll in transgenic plant leaves were significantly decreased (Figure 3C). The biotrophic pathogens growth assay suggested that the number of *P. syringae* pathogen in leaves of transgenic plants (L2 and L9) was significantly more than that in WT, while there also were slightly more *P. syringae* pathogens in L4 and L6 than WT (Figure 3D). Gene expression levels assay shown that most marker genes in the SA signaling pathway, including *PR1*, *PR2*, and *NPR1*, were dramatically decreased in plants overexpressed *PtrWRKY18* or *PtrWRKY35* (Figure 3E). However, the expression level of *PR5* was up-regulated in plants overexpressing *PtrWRKY18*, but significantly down-regulated in plants overexpressing *PtrWRKY35* (Figure 3E). This may be as a result of the two genes functioning through not exactly the same manner.

These results indicated that overexpression of *PtrWRKY18* and *PtrWRKY35* led to a decrease of *P. syringae* resistance in Arabidopsis by downregulating the expression levels of most genes in the SA signaling pathway.

### 2.4. Overexpression of PtrWRKY18 and PtrWRKY35 Reduced Sensitivity to ABA in Transgenic Arabidopsis

As an important component of plant signaling pathways, the hormone ABA acts as a key signal for regulating a range of plant physiological processes, such as germination, seedling growth, root development, stomatal regulation, and defense to osmotic stress [24,25]. From the results of Figure 1B and 1C, we can learn that the expression levels of *PtrWRKY18* and *PtrWRKY35* are downregulated after ABA treatment. Therefore, we speculated that *PtrWRKY18* and *PtrWRKY35* might play roles in the ABA-independent signaling pathway.

To test the hypothesis, the seeds of WT and transgenic plants were treated with ABA during seedling development to observe their response to ABA. Compared with WT plants, the seeds of transgenic plants were germinated earlier and had significantly higher germination rates under the same concentration of ABA treatment (Figure 4A). There was no significant difference in the germination process (data not shown) and root length (Figure 4B,C) between seeds of transgenic plants and WT in the absence of ABA. Although ABA treatment severely inhibited the root development of WT plants, the root length of transgenic plants was obviously longer than that of WT (Figure 4B,C). These results provide clues to the reduced sensitivity of ABA in *PtrWRKY18* or *PtrWRKY35* overexpressing plants.

### 2.5. Overexpression of PtrWRKY18 and PtrWRKY35 Reduced Drought Tolerance in Arabidopsis

In many plants, endogenous ABA levels dramatically accumulate in conditions of osmotic stresses, such as high salinity and drought [26,27]. The increased ABA content activates the expression of downstream transcription factors to regulate various ABA-responsive genes, so that plants can respond to osmotic stress through closing stoma, reducing transpiration, and so on [28,29]. 

On the basis of the phenotype that transgenic plants were less sensitive to ABA (Figure 4), we hypothesized that overexpression *PtrWRKY18* or *PtrWRKY35* may reduce the tolerance of plants to osmotic stresses, and NaCl and drought treatment were used to confirm our speculation. Compared with WT, the *PtrWRKY18* or *PtrWRKY35* overexpression lines were more sensitive to both high salinity and drought stress, especially the drought stress (Figure 5A). After drought treatment, transgenic plants had excessive loss of cellular water and appeared on more severe defect symptoms (Figure 5A). Fewer plants in transgenic plants can be revived after re-watering than the WT plants (Figure 5B). As a critical chemical messenger for osmotic response, ABA has been brought to the central stage of variation of stomatal aperture [30]. Stomata opening and closure of leaves observation showed that, for the stomatal aperture, there was no significant difference between the leaves of WT and transgenic plants in normal conditions (Figure 5C). With ABA treatment, the leaves of *PtrWRKY18* or *PtrWRKY35* overexpressing plants showed a larger stomatal opening than that of WT (Figure 5C). The length/width ratio of stomatal pores was usually used as an indicator of stomatal aperture to analyze the stomatal aperture in ABA-mediated drought response [31]. The statistical results showed that the guard cell of leaves from transgenic plants had a significantly larger stomatal aperture than that of WT (Figure 5D). These results suggest that *PtrWRKY18* and *PtrWRKY35* participated in the ABA-mediated drought response, and reduced the plants’ drought tolerance via affecting the ABA sensitivity.

It has been reported that *RD22* and *RD29A* are stress-response genes, and could be induced by salt, drought, and ABA. *ABO3* mediates the drought tolerance of Arabidopsis through regulating the expression of downstream genes [13]. *ABI4* and *ABI5*, two ABA-insensitive genes, are inhibited by *AtWRKY18*, *AtWRKY40*, and *AtWRKY60* [16]. *DREB1A* also plays a role in drought defense [32]. Therefore, qRT-PCR analysis (Figure 6) was used to research the mechanism of ABA-mediated drought response. As showed in Figure 6, overexpressing *PtrWRKY18* or *PtrWRKY35* in Arabidopsis down-regulated the expression levels of *RD29A*, *ABO3*, *ABI4*, *ABI5*, and *DREB1A*. The results further demonstrated that *PtrWRKY18* and *PtrWRKY35* were involved in the signal pathway of ABA-mediated drought response. Interestingly, *RD22*, which was the marker gene of salt, drought, and ABA stresses, displayed no significant changes in transgenic plants.

In short, *PtrWRKY18* and *PtrWRKY35* can be induced by various stresses, and have a variety of functions in response to environmental stress. Heterologous overexpression of *PtrWRKY18* and *PtrWRKY35* in Arabidopsis not only leads to enhanced JA-induced defense against necrotrophic pathogens and weakened SA-induced defense against biotrophic pathogen, but also affects drought tolerance via the ABA-independent signaling pathway.

## 3. Discussion 

WRKYs have been widely reported to play key and various roles in regulating plant growth and development, including somatic embryogenesis [33], seed coat pigmentation and development [34,35], trichome development [36], leaf senescence [37,38], as well as various abiotic and biotic stress responses [39,40]. Recently, most of the studies about WRKYs are focused on abiotic and biotic stress responses, especially at the transcriptional level [41,42,43]. A great deal of results demonstrate that WRKYs are key regulators in basal defense responses of many plant species, such as Arabidopsis, rice, strawberry, grapevine, poplar, and tobacco [44,45,46,47,48,49,50]. Meanwhile, many WRKY proteins are involved in SA-mediated defense against biotrophic pathogens. In Arabidopsis, 49 members of 72 WRKYs can be induced by SA treatment or significantly regulated through pathogenic infections [51]. *AtWRKY70* has been demonstrated to be an activator of the SA signal pathway to reduce resistance to *Alternaria brassicicola* and enhance resistance to *Erysiphe cichoracerum* [52,53]. JA and SA often play antagonistically roles in defense responses, and some WRKY proteins also take part in JA-mediated defense against herbivorous insects or necrotrophic pathogens. *CaWRKY27* and *CaWRKY40* act as positive regulators in tobacco resistance against *Ralstonia solanacearum* through the regulation of SA-, JA-, and ethylene-mediated signaling pathways [54,55]. In Arabidopsis, overexpression of *AtMYB44* can reduce the plant defense response to the necrotrophic pathogen *Alternaria brassicicola* and enhance resistance to the biotrophic pathogen *Pst DC3000* via regulating *AtWRKY70* expression and modulating antagonistic interaction between the SA and JA signaling pathway [56]. It has also been proven that the double mutants of *Atwrky18*/*Atwrky40* and *Atwrky18/Atwrky60* showed a higher resistance to *P. syringae*, but were more susceptible to *B.cinerea* [15,16].

Recently, growing evidence shows that WRKYs are also involved in the ABA signaling pathway [29,57,58]. In Arabidopsis, *AtWRKY18* and *AtWRKY60* have positive effects on plant ABA sensitivity, and inhibit seed germination and root growth, whereas *AtWRKY40* has a negative effect on plant ABA sensitivity and promotes the seed germination and root growth [15]. Constitutive expression of *GhWRKY17* in tobacco remarkably reduced plant drought and salt tolerance, and enhanced plant ABA sensitivity to inhibit seed germination and root growth [58].

### 3.1. SA, JA, and ABA All Play Important Roles in Regulating the Expression of PtrWRKY18 and PtrWRKY35

In our previous studies, two poplar WRKY transcription factors, *PtrWRKY18* and *PtrWRKY35*, have been identified and isolated [22]. They have been demonstrated to be the homologous genes of *AtWRKY18*, *AtWRKY40*, and *AtWRKY60*, and can activate pathogenesis-related genes to increase resistance to the biotrophic pathogen *Melampsora* via SA-mediated signal pathway in poplar [22]. Their homologous genes in Arabidopsis also have vital effects on plant ABA sensitivity and drought tolerance, and inhibit seed germination and root growth [15]. Therefore, we have a hypothesis that *PtrWRKY18* and *PtrWRKY35* are multifunctional transcription factors, and participate extensively in plant biotic and abiotic stress responses.

To test the hypothesis, various treatments (SA, JA, *P. syringae*, *B. cinerea*, and ABA) were applied to the leaves of transgenic Arabidopsis containing GUS report genes. Expression levels of *PtrWRKY18* and *PtrWRKY35* were significantly up-regulated under the treatments of SA, JA, *P. syringae*, and *B. cinerea* and down-regulated under the ABA treatment (Figure 1). The results suggested that *PtrWRKY18* and *PtrWRKY35* were regulated by multiple hormones.

### 3.2. PtrWRKY18 and PtrWRKY35 Play Antagonistic Roles in JA and SA Signaling Pathway

To study the versatility of *PtrWRKY18* and *PtrWRKY35*, we heterologous overexpressed them in Arabidopsis. *B. cinerea* was introduced to activate JA-mediated defense against necrotrophic pathogens. As expected, overexpression of *PtrWRKY18* or *PtrWRKY35* could activate pathogenesis-related genes to increase resistance to *B. cinerea* in Arabidopsis (Figure 2).

To investigate the effects of heterologous overexpression of *PtrWRKY18* or *PtrWRKY35* on the SA signaling pathway, the transgenic Arabidopsis were treated with *P. syringae*. Surprisingly, overexpression of them in Arabidopsis inhibited the expression of marker genes in the SA signaling pathway, thereby reducing plant resistance to *P. syringae* (Figure 3).

Those results suggest that heterologous expression of *PtrWRKY18* or *PtrWRKY35* in Arabidopsis activated the JA signaling pathway (Figure 2), but inhibited the SA signaling pathway (Figure 3). The results were consistent with antagonistic effects of SA and JA in many aspects of plant growth and development, including defense responses [24].

In our previous study, we demonstrated that *PtrWRKY18* or *PtrWRKY35* played positive roles in the *P. tomentosa* SA signaling pathway [22]. Meanwhile, we have not discovered their function in the JA signaling pathway of *P. tomentosa*. Combined with the research in this paper, we can conclude that the distinct functions of *PtrWRKY18* and *PtrWRKY35* in Arabidopsis and poplar may owing to species difference.

### 3.3. PtrWRKY18 and PtrWRKY35 Play Roles in ABA Signal Perception, and Can Regulate the Tolerance to Osmotic Stress through Stomatal Movement

Generally, ABA acts as an important hormone to deal with various biotic and abiotic stresses. High salinity and drought stimulation can promote its accumulation [25,26]. In this study, we found that the expression of *PtrWRKY18* and *PtrWRKY35* was inhibited by ABA signals and that, compared with WT, the germination rate and root development of Arabidopsis were suppressed (Figure 4). The results suggest that they may negatively regulate ABA signaling.

We have shown that overexpression of *PtrWRKY18* and *PtrWRKY35* remarkably reduced the tolerance of Arabidopsis to high salinity and drought treatment (Figure 5A,B). As a major indicator of ABA-mediated osmotic stress response, the stomatal aperture of guard cells from transgenic plants was larger than that of WT under ABA treatment (Figure 5C,D). Some key transcription factors in ABA pathway were also down-regulated in transgenic plants (Figure 6). The results told us that *PtrWRKY18* and *PtrWRKY35* indeed played negative roles in the ABA-dependent signaling pathway.

Taken together, *PtrWRKY18* and *PtrWRKY35*, with similarities to *AtWRKY18*, *AtWRKY40*, and *AtWRKY60*, were multifunctional transcription factors in plant resistance. In plant immunity, *PtrWRKY18* and *PtrWRKY35* not only play important roles in the SA-mediated pathway [22], but also function in regulating the JA signaling pathway. Meanwhile, they have a negative effect on ABA sensitivity, and reduce plants’ tolerance to osmotic stress.

To better understand the respective functions of *PtrWRKY18* and *PtrWRKY35*, overexpression of *PtrWRKY18* and *PtrWRKY35* in *AtWRKY18*, *AtWRKY40*, and *AtWRKY60* single, double, or triple mutants may be viable methods. A more detailed functional analysis of *PtrWRKY18* and *PtrWRKY35* will help us improve poplar trees through genetic engineering techniques, and cultivate various poplar varieties that can adapt to extreme environments.

## 4. Materials and Methods

### 4.1. Plant Material and Treatments

As described in our previous study (Jiang et al. 2007), the full-length coding sequences of *WRKY18* and *WRKY35* were amplified from the cDNA that was reversely transcribed from the mRNA of *P. trichocarpa*. The amplified cDNA fragments were subsequently constructed in the pCXSN vector under the control of the *35S* promoter for overexpression. The resulting constructs were genetically transformed into Arabidopsis using the method of agrobacterium-mediated floral dip. The positive transgenic plants of T1 generation were screened out by hygromycin resistance and PCR genotyping, and pollinated for T2 generation. The produced T2 seeds were germinated on hygromycin-containing MS medium, and calculated for the ratio of survival/death (3:1) to identify the transgenic plants of single insertion. The single copy-inserted transgenic plants were pollinated for T3 generation. The T3 seeds were germinated on the medium supplemented with hygromycin, and the lines, the seedlings of which all survived, were considered as the homozygotes. The highly expressed transgenic lines were detected by RT-PCR using the primer used previously [22].

The seeds of WT Col-0, homozygous *PtrWRKY18*, and *PtrWRKY35* overexpressing lines were kept at 4 °C for 3 days before being placed on MS medium [59] supplemented with 3% (*w*/*v*) sucrose. After 10 days of germination on MS plates, the seedlings were transferred into soil and developed in a growth incubator at 22 °C under long-day conditions (16 h light/8 h dark), with 80% relative humidity for further analysis.

For hormonal treatments, SA (5 mM in water), JA [1 mM in 0.1% (*v*/*v*) ethanol], and ABA [25 μM in 0.1% (*v*/*v*) ethanol] solutions were sprayed on whole plants, respectively. The water or the solution of 0.1% (*v*/*v*) ethanol without any hormones was used as the mock control. Each treatment was performed for more than four biological replicates. The treated plants were covered with a transparent film sheet for 24 h. Then, leaves were detached for GUS staining. Inoculation of *Botrytis cinerea* (*B. cinerea*) and *Pseudomonas syringae pv. tomato DC3000* (*P. syringae*) was performed as described previously [60].

In order to observe the difference in sensitivity to ABA during seed germination, seeds were evenly placed on MS medium supplemented with 3% (*w*/*v*) sucrose and 0.3 μM ABA. Statistical germination rates were calculated from 1 to 9 days after the earliest germination (DAG). For root length statistics, seeds were placed on square petri dish containing normal MS medium supplemented with 3% (*w*/*v*) sucrose and MS medium supplemented with 3% (*w*/*v*) sucrose and 10 μM ABA, respectively. Root lengths were measured after 8 days of vertical culture.

For NaCl and drought treatment, WT and transgenic plants were germinated simultaneously on the culture dishes and then planted in soil. After 3 weeks of growth, those plants were cultivated in 300 mM NaCl treatment for 7 days or natural drought conditions (water was withheld) for 14 days, respectively. For the death ratio, those plants that suffered drought were watered again for 7 days. Each experiment was repeated three times and at least 10 plants from the individual lines were used in each experiment.

### 4.2. GUS Staining

The 1500 bp promoter fragments of *PtrWRKY18* and *PtrWRKY35* were amplified from the genomic DNA of *P. trichocarpa*, and ligated into pCXGUS-P vector to drive the *GUS* (β-glucuronidase) reporter gene to obtain the vectors of *PtrWRKY18pro: GUS* and *PtrWRKY35pro: GUS*, respectively. The constructs were transferred into *Agrobacterium tumefaciens* GV3101, and transformed into Arabidopsis using the floral dip method. Single-inserted positive plants were selected according to the method mentioned above. Two-week-old homozygous transgenic plants of T3 generation harboring a single copy of insertion were treated with hormones or inoculated with fungal pathogens. Then, the leaves were detached to detect GUS activity via histochemical staining. GUS staining was performed in X-Gluc solution (2mM X-Gluc, 0.1 M sodium phosphate buffer (pH 7), 2 mM K_4_Fe(CN)_6_, 2 mM KNaPO4 (pH 7) 10 mM EDTA, 2 mM K_3_Fe(CN)_6_, and 0.2% Triton X-100) under a tubes at 37 °C for 4 h in the dark. Chlorophyll was removed using 70% (*v*/*v*) ethanol. At least five leaves independent transgenic lines of *WRKY18pro: GUS* or *WRKY35pro: GUS* as biological replicates for each treatment were used for GUS staining.

### 4.3. Chlorophyll Content Detection

Extraction and measurement of chlorophyll were performed as previously described [60]. Briefly, 0.2 g of leaves was fully grinded with a small amount of calcium carbonate powder and quartz sand in 3 mL acetone. All homogenates were combined in 80% (*v*/*v*) acetone and filtered with filter paper. Finally, the filtrate was added to 100 mL with 80% (*v*/*v*) acetone, and then the absorbance of supernatant at 663 (A663) and 645 (A645) nm was determined using UV/visible spectrophotometer Model DU800 (Shimadzu Corporation, Kyoto, Japan). The total chlorophyll (C) content was calculated using the formula below: C (mg/g) = (20.2 A645 + 8.02 A663)/2. The measurements were repeated for three biological replicates of each transgenic line.

### 4.4. Growth State Detection of P. syringae in Plants

The growth situation of *P. syringae* in plants was indicated by the number of colonies in the leaves. The injected leaves of Arabidopsis were diluted and then spread on KB (King’s B) medium, and the number of *P. syringae* colonies on the flat plate was counted after three days.

### 4.5. Stomatal Movement Assay Response to ABA

Rosette leaves of two-week-old plants were floated in buffer for total opening of stomatal containing 10 mM MES-Tris, 50 mM KCl, pH 6.5, and exposed to light for 2.5 h. Subsequently, ABA was added to the solution up to 25 μM. After ABA treatment for 2.5 h, stomatal length and width were measured under the microscope, and the ratio of stomatal width to length was used as an indicator of stomatal opening. Thirty biological replicates were conducted.

### 4.6. Gene Expression Analysis

For quantitative real time PCR (qRT-PCR) analysis, total RNA from fresh tissues was extracted using RNA RNeasy Plant Mini Kit (Qiagen, Hilden, Germany) and treated with RNase-free DNase (TaKaRA, Dalian, China). Samples from at least three plants were pooled for analysis. The quality or integrity of RNA was checked by agarose gel electrophoresis and P100 spectrophotometer (Pultton, Ann Arbor, MI, USA). The criteria of high-quality total RNA include the following: (1) sharp distinct 28S and 18S rRNA bands, with the 28S band approximately twice as intense as the 18S band; (2) the value of D260/OD280 between 1.9 and 2.0; and (3) no detected genomic DNA band. The qualified RNA samples were reversely transcribed using RT-AMV (Avian Myeloblastosis Virus) transcriptase (TaKaRa, Dalian, China). PCR amplification was performed for CDS of a random gene containing an intron to exclude DNA contamination. Subsequently, qRT-PCR was performed in a volume of 25 μL containing 12.5 μL of SYBR Premix ExTaq TM (TaKaRa, Dalian, China). *ACTIN2* rRNA was used as an internal control. Five biological replicates of each sample and three technical replicates of each biological experiment were conducted. Primers used for qRT-PCR were listed in Appendix A.

## 5. Conclusions

In the research, we isolated two WRKY TF genes *PtrWRKY18* and *PtrWRKY35* from *Populus trichocarpa* and overexpressed them in Arabidopsis. The results indicated that they play roles in antagonistic regulation on pathogen resistance and abiotic stress tolerance via variable JA, SA, and ABA pathways. However, the molecular mechanism and crosstalk of hormonal pathways remain unclear. Future work is needed to better understand how WRKY transcription factors participate in pathogen resistance and abiotic stress tolerance via several hormonal pathways together.

## Figures and Tables

**Figure 1 ijms-21-05440-f001:**
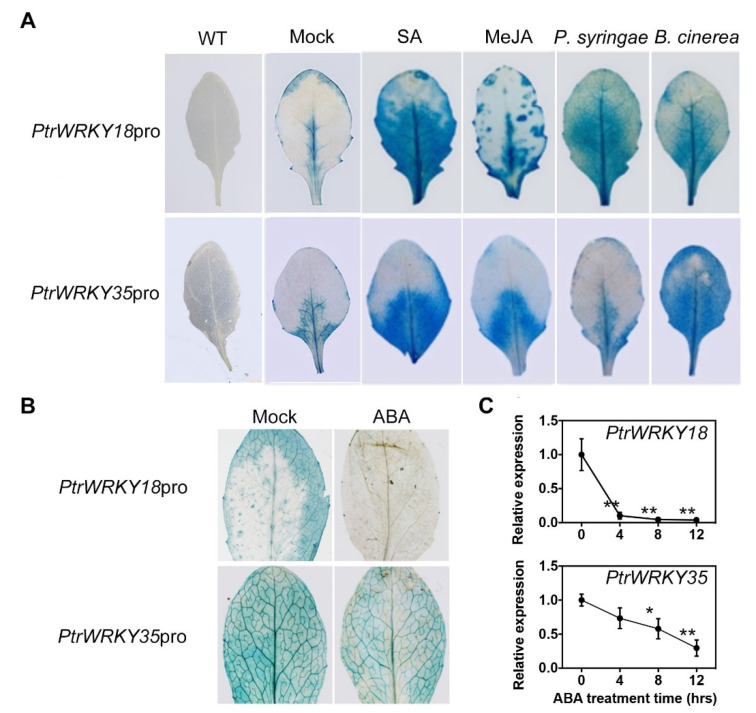
Expression analysis of *PtrWRKY18* and *PtrWRKY35*. (**A**) GUS (β-glucuronidase) staining of leaves of transgenic Arabidopsis overexpressing *PtrWRKY18pro: GUS* and *PtrWRKY35pro: GUS* after treated with SA, JA, *B. cinerea*, and *P. syringae*. Wild-type (WT) leaves and the transgenic leaves under mock treatment were used as negative control. (**B**) GUS staining of leaves of transgenic Arabidopsis overexpressing *PtrWRKY18:GUS* and *PtrWRKY35:GUS* after being treated with ABA for 4 h. (**C**) Time-course quantification of GUS activities under abscisic acid (ABA) treatment. Five biological replicates were determined for each timepoint. The values for 0 h were normalized to 1, and bars represent SD. Asterisks indicate significant differences with respect to the value for 0 h ((Student’s *t*-test: * *p* < 0.05; ** *p* < 0.01; *n* = 5). MejA, methyl JA; SA, salicylic acid.

**Figure 2 ijms-21-05440-f002:**
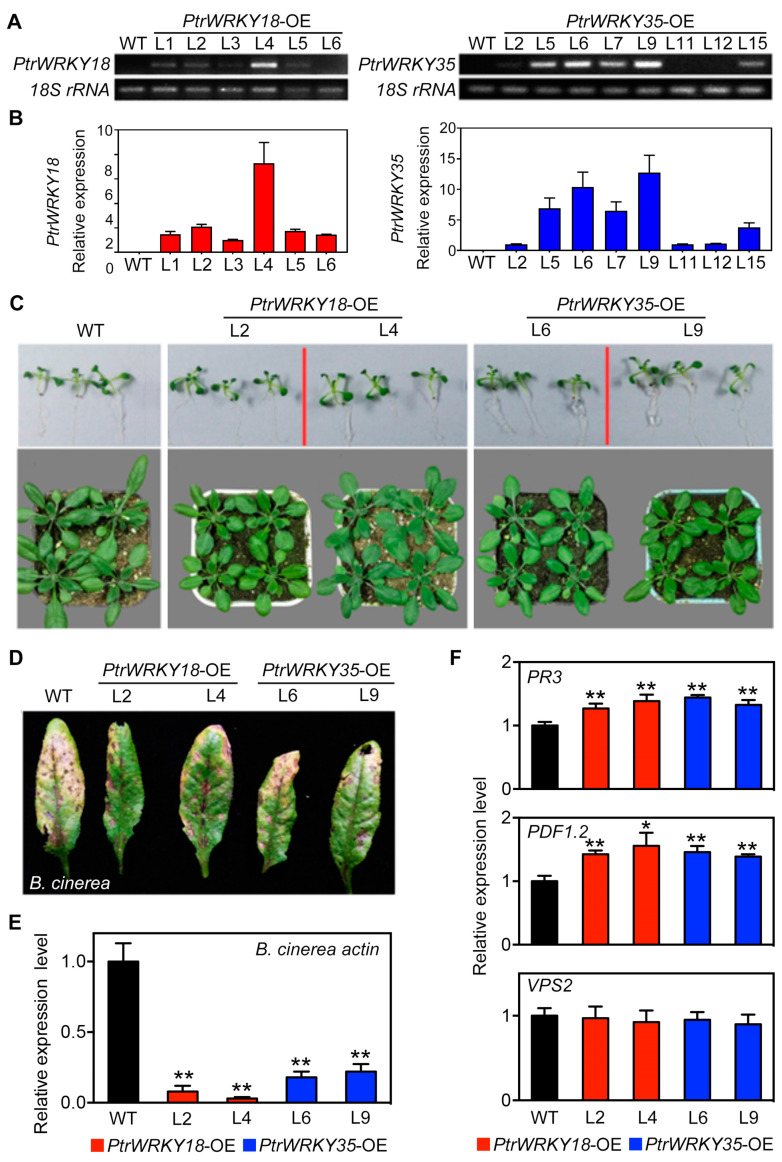
Constitutive expressing *PtrWRKY18* or *PtrWRKY35* enhanced the *Botrytis cinerea* tolerance of Arabidopsis. (**A**,**B**) Semi real-time quantitative (qRT)-PCR (**A**) and qRT-PCR (**B**) analysis of transgenic Arabidopsis lines overexpressing *PtrWRKY18* or *PtrWRKY35*. (**C**) Phenotype observation of two-week-old and one-month-old transgenic and WT plants. (**D**) Leaves phenotypes of WT, *PtrWRKY18*, and *PtrWRKY35* overexpressed plants after *B. cinerea* inoculation. (**E**) Expression levels of *Bcactin* (*B. cinerea* actin gene) in infected plants. (**F**) Expression levels analysis of JA response marker genes in WT and plants overexpressing *PtrWRKY18* and *PtrWRKY35*. *PR3*: *pathogenesis-related gene 3*; *PDF1.2*: *plant defensin genev1.2*; *VSP2*: *vegetative storage protein gene 2*. For (**E**,**F**), average values from three biological replicates were shown. The error bars are used to represent SD (* *p* < 0.05; ** *p* < 0.01).

**Figure 3 ijms-21-05440-f003:**
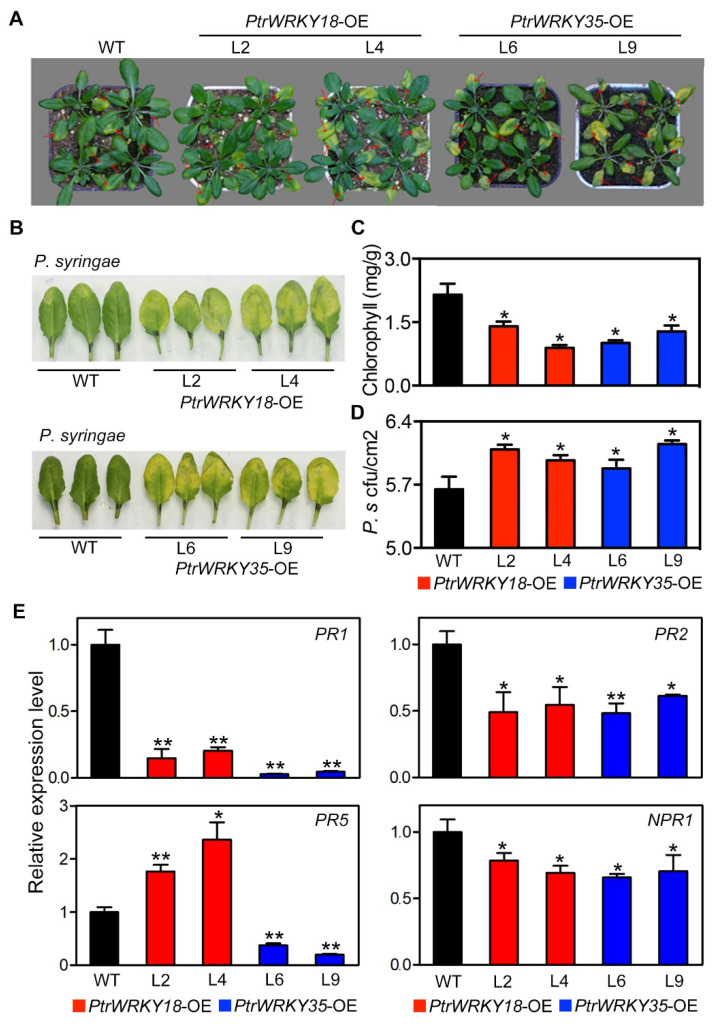
The Arabidopsis overexpressing *PtrWRKY18* or *PtrWRKY35* had decreased resistance to *P. syringae*. (**A**) Phenotype of WT and transgenic plants overexpressing *PtrWRKY18* or *PtrWRKY35* infected with pathogen *P. syringae*. (**B**) Leaves phenotypes of WT, *PtrWRKY18*, and *PtrWRKY35* transgenic plants after *P. syringae* inoculation. (**C**) Quantification of total chlorophyll content in WT and transgenic plant infected with *P. syringae*. (* *p* < 0.05; *n* = 3). (**D**) Growth of *P. syringae* in WT and transgenic plants after inoculation (* *p* < 0.05; *n* = 3). For **C** and **D**, each experiment was carried out with 20 plants and the experiment was repeated three times. (**E**) Expression analysis of maker genes involved in SA signaling pathway. *PR1/2/5*: *pathogenesis-related genes 1/2/5. NPR1: non-expresser of PR genes 1*. Three repeats of biological replicates were performed for each gene, and each experiment contained three technical replicates. Values are means with SD (* *p* < 0.05; ** *p* < 0.01).

**Figure 4 ijms-21-05440-f004:**
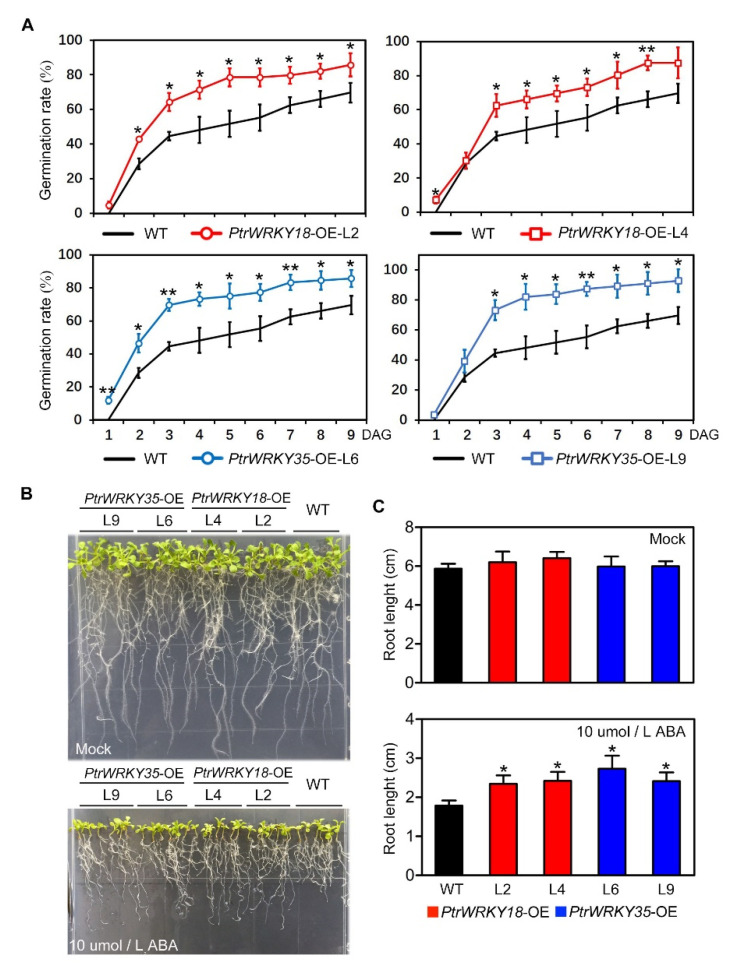
The Arabidopsis overexpressed *PtrWRKY18* or *PtrWRKY35* showed the ABA-insensitive phenotype. (**A**) Quantitative comparison of seeds’ germination ratio between WT and plants overexpressing *PtrWRKY18* or *PtrWRKY35* on MS medium supplemented with ABA. Each time, the statistics contained 28 plants and three replicates were performed. The same WT line was used as control. The error bars were used to represent SD (* *p* < 0.05; ** *p* < 0.01). (**B**) Phenotype of root of WT and transgenic plants in MS and MS containing ABA. (**C**) Root length statistics of WT and transgenic plants in MS and MS containing ABA. Fifteen plants were used for each statistic, and three independent experiments were performed. SD is represented by error bars (* *p* < 0.05).

**Figure 5 ijms-21-05440-f005:**
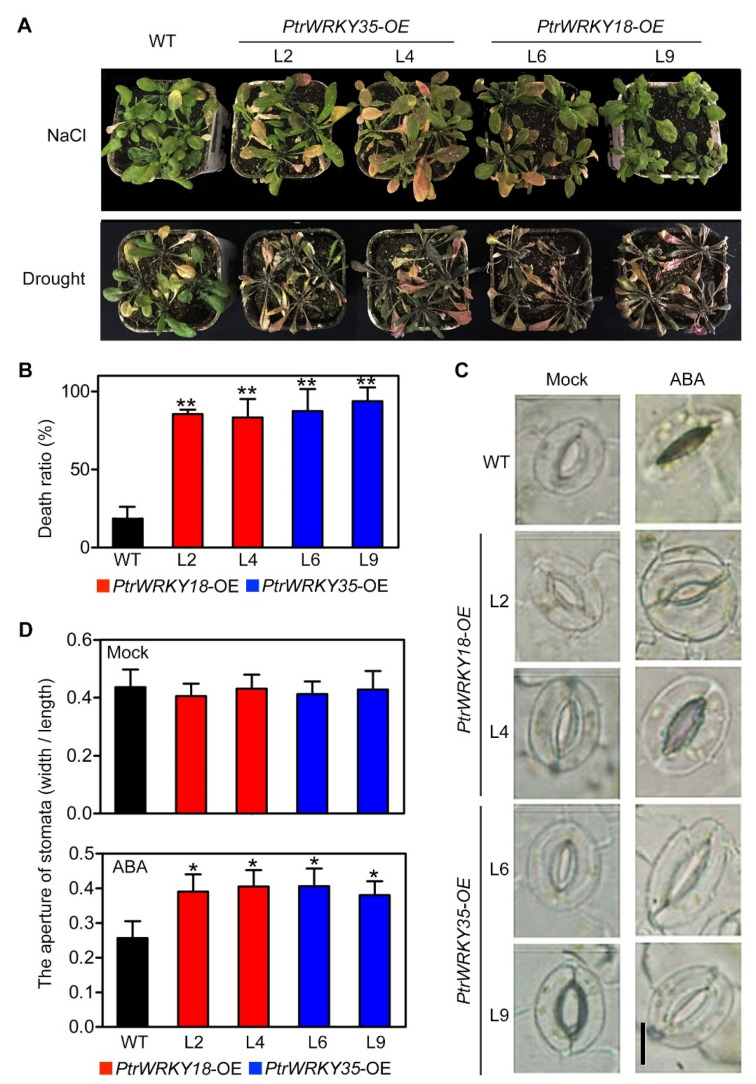
*PtrWRKY18* and *PtrWRKY35* negatively regulate plant drought defense in Arabidopsis. (**A**) Phenotypes of WT, *PtrWRKY18*, and *PtrWRKY35* overexpressing plants treated with NaCl or natural drought. (**B**) Death ratio of WT and transgenic plants in (**A**) after re-watering. Thirty plants were used for each experiment, and three independent experiments were done. Values are means with SD (** *p* < 0.01). (**C**) Microscopic observation of guard cells of WT and Arabidopsis overexpressing *PtrWRKY18* or *PtrWRKY35* before and after ABA treatment, bars = 10 μm. (**D**) Stomatal aperture (the ratio of stomatal width to length) measurements of WT and transgenic plants before and after ABA treatment. In each experiment, 40 stomata from different plants were conducted. Three independent experiments were performed. Error bars are used to indicate SD (* *p* < 0.05).

**Figure 6 ijms-21-05440-f006:**
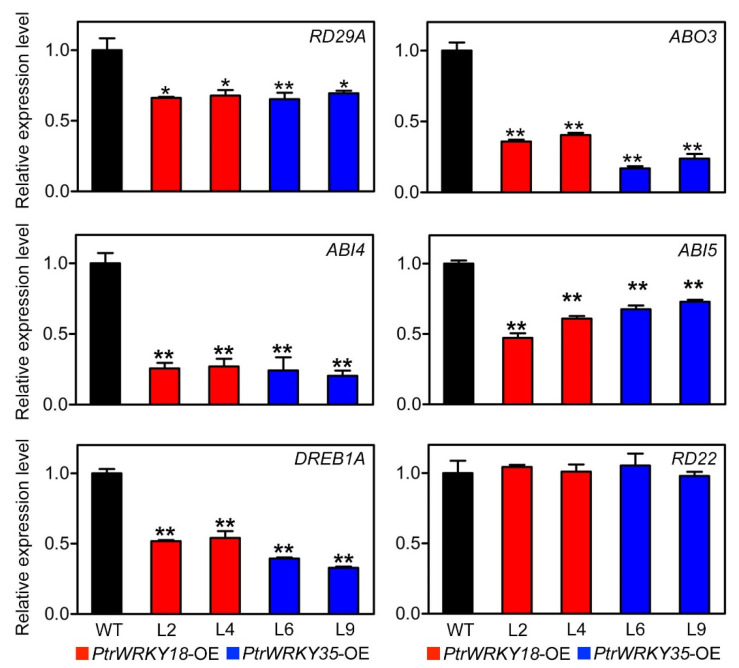
Expression level of ABA-related genes in WT and plants overexpressing *PtrWRKY18* or *PtrWRKY35*. *RD29A*: *responsive to desiccation 29A*, *ABO3*: *ABA overly sensitive 3*, *ABI4/5*: *ABA insensitive 4/5*, *DERB1A*: *dehydration response element B1A*, *RD22*: *responsive to dehydration 22*. Three repeats of biological replicates were performed for each gene, and each experiment contained three technical replicates. Actin was used as a reference for normalization, error bars represent SD (* *p* < 0.05; ** *p* < 0.01).

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
