# Peer review of "Heterologous Expression of Poplar *WRKY18/35* Paralogs in Arabidopsis Reveals Their Antagonistic Regulation on Pathogen Resistance and Abiotic Stress Tolerance via Variable Hormonal Pathways"

_ijms, 2020, doi:10.3390/ijms21155440_

Round 1

Reviewer 1 Report

Presented for review manuscript entitled "Heterologous expression of poplar WRKY18/35 paralogs in Arabidopsis reveals their antagonistic regulation on pathogen resistance and abiotic stress tolerance via variable hormonal pathways" showed well-prepared and presented research. Below you are going to find the minor comments:

  • authors should write in more detail what was a mock for each hormone treatment when the hormones are prepared in different solvents
  • in section "chlorophyll content detection" authors should shortly describe the method. In a few words should write how it was performed
  • "section gene expression analysis" how was checked RNA purity? How authors checked that sample is not contaminated by DNA after DNase treatment?
  • Generally, authors should check the level of hormones in the transgenic plants exposed to hormones and pathogens. Moreover, it would be nice to show how looks expression level of crucial genes involved in biosynthesis and degradation of tested hormones in transgenic plants exposed to hormones and pathogens. Those results would give a nice picture of WRKY regulation in antagonistic regulation on pathogen resistance and abiotic stress tolerance via variable JA, SA, and ABA pathways

To summing up I accept this manuscript for publication in IJMS after minor revision.

Author Response

Presented for review manuscript entitled "Heterologous expression of poplar WRKY18/35 paralogs in Arabidopsis reveals their antagonistic regulation on pathogen resistance and abiotic stress tolerance via variable hormonal pathways" showed well-prepared and presented research. Below you are going to find the minor comments:

  • 1. authors should write in more detail what was a mock for each hormone treatment when the hormones are prepared in different solvents.

Response 1: Thanks for the suggestion. SA (5mM in water), JA [1 mM in 0.1% (v/v) ethanol] and ABA [25 μM in 0.1% (v/v) ethanol] solutions were sprayed on whole plants, respectively. The water or the solution of 0.1% (v/v) ethanol without any hormones was used as the mock control. Since the mock controls by water or the solution of 0.1% (v/v) ethanol similarly exhibited very weak GUS staining, the mock presented in Figure 1A was the solution of 0.1% (v/v) ethanol. The detailed description has been added to the “Materials and Methods” and the legend of Figure 1 in the manuscript.

  • 2. in section "chlorophyll content detection" authors should shortly describe the method. In a few words should write how it was performed.

Response 2: Thanks for the suggestion. The experimental process of chlorophyll content assays was described in the “Materials and Methods”, as it is “The extraction and measurement of chlorophyll was performed as previously described methods for measurement of chlorophyll were described previously by Jiang et al [63]. Briefly, 0.2 g of leaves was fully grinded with a small amount of calcium carbonate powder and quartz sand in 3 ml acetone. All homogenates were combined in 80% (v/v) acetone and filtered with filter paper. Finally, the filtrate was added to 100mL with 80% (v/v) acetone, and then the absorbance of supernatant at 663 (A663) and 645 (A645) nm was determined using UV-visible spectrophotometer Model DU800 (Shimadzu Corporation, Kyoto, Japan). The total chlorophyll (C) content was calculated using the formula below: C (mg/ g) = (20.2* A645 + 8.02* A663)/ 2. The measurements were repeated for three biological replicates of each transgenic line.”

  • 3. "section gene expression analysis" how was checked RNA purity? How authors checked that sample is not contaminated by DNA after DNase treatment?

Response 3: Thanks for the comment. For qRT-PCR analysis, total RNA from fresh tissues was extracted using RNA RNeasy Plant Mini Kit (Qiagen, Germany) and treated with RNase-free DNase (TaKaRA, Dalian, China). The quality or integrity of RNA was checked by agarose gel electrophoresis and Nano Drop spectrophotometer (Pultton, American). The criteria of high-quality total RNA include: 1) sharp distinct 28S and 18S rRNA bands, with the 28S band approximately twice as intense as the 18S band; 2) the value of OD260 / OD280 between 1.9-2.0; 3) no detected genomic DNA band. The qualified RNA samples were reversely transcribed using RT-AMV transcriptase (TaKaRa, Dalian, China). PCR amplification was performed for CDS of a random gene containing an intron to exclude DNA contamination. The information has been described in the “Materials and Methods” of the manuscript.

  • 4. Generally, authors should check the level of hormones in the transgenic plants exposed to hormones and pathogens. Moreover, it would be nice to show how looks expression level of crucial genes involved in biosynthesis and degradation of tested hormones in transgenic plants exposed to hormones and pathogens. Those results would give a nice picture of WRKY regulation in antagonistic regulation on pathogen resistance and abiotic stress tolerance via variable JA, SA, and ABA pathways.

Response 4: Thanks for the comment. Increasing evidence has demonstrated WRKYs play vital roles in pathogen defense, and can be are regulated by elicitors such as wounding, SA and JA (Oh et al. 2006; Zheng et al. 2007; Lai et al. 2008; Karim et al. 2015). Moreover, WRKYs were also found to function as components of ABA signaling to drive gene expression. Here we show that the expression of WRKY18 and WRKY35 can be induced by pathogens and various hormones including SA, JA, and ABA, and both WRKYs mediate the expression of SA-, JA-, and ABA-regulated expression of genes involved in stress response. Therefore, these WRKY paralogs function possibly as the downstream key signaling components of these hormones. The quantification of hormone content may not contribute to the elucidation of WRKY18/35-regulated stress tolerance. If necessary, we can quantify the hormone content in the transgenic lines. However, due to the experimental period of material preparation, hormone extraction and quantification, please allow us for more time to complete the experiment.

Reviewer 2 Report

Summary;

The purpose of this manuscript was the investigate the functions of two WRKY family transcription factors from Populus by expressing them in A. thaliana. Two transcription factor genes, PtrWRKY18 and PtrWRKY35 were overexpressed in Arabidopsis to determine their effects on pathogen resistance and abiotic stress tolerance. Initially, the authors made promoter:GUS fusions to determine the impacts of hormone (ABA, MeJA, SA) and pathogen (P. syringae, B. cinera) on promoter activity. They found that ABA treatment led to decreasing promoter activity over time, while other treatments all led to higher than mock levels of GUS staining. Next, the authors created overexpression constructs and selected two independent events per construct to investigate gene expression levels under abiotic and biotic stress conditions and hormone treatments. The overexpression events showed decreased colonization and leaf damage by B. cinerae, perhaps due to increased expression of defense pathway genes. By contrast, overexpression events were mor sensitive to P. syringae, with worse chlorosis than WT plants, but with minimal increase in bacterial growth. Overexpression of PtrWRKY18 and PtrWRKY35 led to inhibition of SA signaling pathway, but activation of the JA pathway. PtrWRKY18 and PtrWRKY35 were determined to reduce plant tolerance to drought and salinity when overexpressed. The authors concluded that the two genes play important roles in regulating pathogen resistance and abiotic stress. This paper presents a lot of quality data investigating the roles of these two transcription factors. It would be strengthened by addressing the following concerns.

Main concerns

The authors do a very thorough investigation of the roles of these transcription factors in A. thaliana. The presentation would be strengthened by a stronger connection/discussion of their roles in poplar. How many WRKY genes are in poplar? Is there evidence of increased expression during biotic or abiotic stress of poplar? Given that poplar trees are perennials, these trees may face such challenges on a reoccurring basis. Please present a stronger connection to the plant of origin.

1. Some of the methods require additional information. Specifically, the sizes of the promoters used for the GUS fusions should be included, as should information regarding the overexpression constructs. Were these 35S-overexpression constructs? Genomic or cDNA clones? How did the authors verify that the events were homozygous and single-insertion?

2. Figure 1 is missing a control as there are no WT samples included. Given that the promoter:GUS mock samples show some level of blue coloration it would be informative to compare them to a true negative control. In addition, it is unclear which timepoint is being shown in panel B, please add these data. It would also be good to know the ages of the plants, the generation used, and the numbers of independent events examined.

3. Figure 3 shows some unusual leaf morphology present in the overexpression lines, the leaves are noticeably round. If these images are representative then it would be useful to quantify this phenotype as ABA has a known role in regulating leaf shape.

Minor concerns

4. Did the authors verify that the overexpression lines have leaves of a similar density (mg/cm2) as control plants? My concern is that the altered leaf shape of the overexpression lines would lead to an altered density, which could skew the cfu/cm2 data presented.

5. The graphs show in figure 4 all appear to have the same WT line. Is this true? It makes sense to break out the data into four graphs such that the four experimental datasets are not overlapped to the point of being unreadable. Please add a comment if the same WT is shown in each panel or not.

5. Drought is misspelled in figure 5 (drougt)

6. The authors do have access to a qPCR machine but opted to use semiQ PCR to select overexpression events. The data would be improved by using QPCR analysis of expression levels. Even if these are not the highest expression levels the analysis of the phenotype are still valid.

Round 2

Reviewer 1 Report

Dear Authors,

Thank you for correcting the manuscript and replying to my comments. In this form, I recommend the manuscript for publication in IJMS. 

Reviewer 2 Report

Thank you for the thorough and comprehensive revisions of your work.  All of the comments and critiques were addressed by the changes.  I have no additional concerns.